# Lipid Polarization during Cytokinesis

**DOI:** 10.3390/cells11243977

**Published:** 2022-12-08

**Authors:** Govind Kunduri, Usha Acharya, Jairaj K. Acharya

**Affiliations:** Cancer and Developmental Biology Laboratory, National Cancer Institute, Frederick, MD 21702, USA

**Keywords:** cytokinesis, sphingolipids, phosphatidylinositol phosphates, lipids, lipid polarization, membrane traffic, male meiotic cytokinesis, multivesicular endosomes, multivesicular bodies, forward trafficking, endocytic recycling, cortical flow, transbilayer coupling, rab GTPase, membrane curvature, membrane bending, cholesterol, phosphatidylethanolamine, triacylglycerols, phosphatidic acid, phosphatidyl serine, phosphatidylinositol, very long chain fatty acids, very long chain polyunsaturated fatty acids, PUFA, disease, sphingolipidoses, lowe syndrome, cataract, aging, cancer, sphingomyelin, ceramide phosphoethanolamine

## Abstract

The plasma membrane of eukaryotic cells is composed of a large number of lipid species that are laterally segregated into functional domains as well as asymmetrically distributed between the outer and inner leaflets. Additionally, the spatial distribution and organization of these lipids dramatically change in response to various cellular states, such as cell division, differentiation, and apoptosis. Division of one cell into two daughter cells is one of the most fundamental requirements for the sustenance of growth in all living organisms. The successful completion of cytokinesis, the final stage of cell division, is critically dependent on the spatial distribution and organization of specific lipids. In this review, we discuss the properties of various lipid species associated with cytokinesis and the mechanisms involved in their polarization, including forward trafficking, endocytic recycling, local synthesis, and cortical flow models. The differences in lipid species requirements and distribution in mitotic vs. male meiotic cells will be discussed. We will concentrate on sphingolipids and phosphatidylinositols because their transbilayer organization and movement may be linked via the cytoskeleton and thus critically regulate various steps of cytokinesis.

## 1. Introduction

Division of one cell into two daughter cells is among the basic and essential functions of cells for their proliferation and growth. Animal cell membranes undergo extensive remodulation in preparation for cell division. At the onset of mitosis, cells shift their shape from being flat at interphase to round at metaphase and elongated ovoid shape at anaphase (Figure 1) [1,2]. Deep invagination of the plasma membrane around the equatorial region with the help of an actomyosin contractile ring in the elongated cell creates a cleavage furrow during cytokinesis, the final stage of cell division (Figure 1). Cleavage furrow ingression, stabilization, and cutting of the membrane (abscission) in the intercellular bridge eventually leads to the separation of nascent daughter cells (Figure 1) [3,4,5,6,7,8]. Precise execution of cytokinesis is essential for genome stability and viability. Since cytokinesis occurs immediately after chromosomal segregation, failure of cytokinesis has been associated with polyploidy, chromosomal instability as well as aneuploidy, features characteristic of many types of cancers [9].

A growing number of studies suggest that successful assembly of the actomyosin ring, cleavage furrow ingression and abscission at the midbody critically depends on lipid trafficking and polarization [4,10]. Lipids with unique biophysical properties have been shown to be enriched at the cleavage furrow, intercellular bridge and midbodies [11,12]. However, the mechanisms by which specific lipids accumulate at the target site and the mechanisms by which they promote cytokinesis remain an active area of research in the field of cytokinesis. Abnormal lipid metabolism affects cytokinesis and has been linked to several human diseases (Box 1). In this review, we will discuss physical and chemical properties of various lipids in promoting high membrane curvature and their distinct distribution during cytokinesis. We will address differences in lipid composition and distribution in mitotic and male meiotic cells. Finally, we will discuss various mechanisms involved in lipid polarization during cytokinesis. We will specifically focus on the role of sphingolipids and phosphatidylinositols in the regulation of cytokinesis. 

Box 1Diseases with cytokinetic failures due to abnormal lipid metabolismAlthough cytokinetic failures are normal and important for certain physiological functions such as differentiation of hepatocytes, cardiomyocytes, vascular smooth muscle, megakaryocytes, and germ cell differentiation, cytokinetic failures in certain tissues induce pathological states such as cancer, certain blood diseases, female infertility, Lowe syndrome, neurofibromatosis type II, sphingolipidoses, and age-related macular degeneration [13]. Here we briefly discuss sphingolipidoses, Lowe syndrome, cataract in age-related diseases and cancer due to their direct association with lipid metabolism.
**Sphingolipidoses:**
Sphingolipidoses are a group of inherited lysosomal storage disorders caused by deficiency of lysosomal enzymes, such as β-galactosylcerebrosidase in Krabbe disease, β-glucocerebrosidase in Gaucher disease, Globotriaosylceramidase in Fabry disease and acid sphingomyelinase in Niemann Pick type A&B [14,15]. Patients deficient in these enzymes show significant accumulation of its immediate sphingolipid precursor such as β-galactosylceramide and its corresponding metabolic deacylated lyso-form, galactosylsphingosine (psychosine) in Krabbe disease. The metabolic origin of psychosine was unknown until recently; Yedda Li et. al. have shown that psychosine was generated through the catabolic deacylation of galactosylceramide by acid ceramidase [16]. One of the characteristic features of Krabbe disease is the presence of multinucleated cells (globoid cells) in the white matter of the brain. Interestingly, psychosine was shown to induce cytokinetic defects and the formation multinucleated giant cells in several mouse and human cell lines in vitro [17]. Cytokinetic defects were also observed when cultured primary human umbilical vein endothelial cells were treated with quasi-pathological concentrations of beta-glucosylsphingosine, a lysosphingolipid that accumulates in Gaucher’s disease [18]. These studies suggested that lysosphingolipids are potent inhibitors of cytokinesis. Although molecular mechanisms are not fully understood, psychosine was shown to accumulate in membrane microdomains and disrupt lipid rafts [19], disperse intracellular membrane vesicles, prevent their accumulation at the cleavage furrow [20], and activate G-protein coupled receptors, T cell death-associated gene 8 (TDAG-8), and inhibit forskolin-driven accumulation of cAMP [17].
**Lowe syndrome:**
Oculocerebrorenal dystrophy, also known as Lowe syndrome, is caused by mutation in the OCRL gene that encodes for inositol 5-phosphatase. In Drosophila, OCRL was shown to mediate PI(4,5)P_2_ homeostasis on endosomal membranes and is required for cytokinesis [21,22]. In the absence of OCRL, PI(4,5)P_2_ was shown to accumulate on endosomal membranes, and as a result, several cleavage furrow-specific proteins are mislocalized to endosomes instead of cleavage furrow, leading to cytokinetic failure [21]. OCRL was shown to be required for remodelling of PI(4,5)P_2_ and F-actin disassembly at the intercellular bridge during cytokinetic abscission [23]. Although cytokinesis failures have not been linked to Lowe syndrome pathologies, cytokinetic abscission defects are observed in Lowe syndrome patient-derived cell lines [23].
**Cataract, senescence, and premature aging:**
Cytokinesis defects result in binucleation and polyploidy, leading to cellular senescence. It was shown that the late onset of cataracts in elderly people is linked to cytokinesis and premature cellular senescence [24,25]. Further, mutations in ESCRT complex subunits CHMP4B and VPS4 are associated with the early onset of cataract in vitro [26,27,28]. Interestingly, Phosphatidylinositol-4-phosphate 3-kinase (PIK3C2A)-null patients show congenital syndromic features resembling premature aging, early onset of cataract and secondary glaucoma [29]. Federico Gulluni et al. recently demonstrated that PIK3C2A localizes to midbody by binding to γ-tubulin and PI(4,5)P_2_. At the midbody, PIK3C2A generates PI(3,4)P_2_ using PI(4)P as a substrate and recruits ESCRT-II/VPS36, which recruit ESCRT-III subunit CHMP4B to the midbody in an ALIX independent pathway to mediate abscission. PI(3,4)P_2_ is required for VPS36 localization to the midbody [10].
**Cancers:**
Cancer is caused by abnormal cells that divide uncontrollably and have the ability to spread to other parts of the body. For offspring with the correct complement of chromosomes, chromosome segregation must be tightly coordinated with cytokinesis. Failure in cytokinesis results in multinucleated polypoid and aneuploid cells that have been historically observed in tumors [9,13]. Although much less is known, dysregulated lipid metabolism is commonly found in cancer cells, suggesting cancer cells require lipids for their rapid divisions [30]. Diacylglycerol (DAG), Sphingosine 1-phosphate (S1P), and ceramides are important signaling second messengers that have been implicated in cancers [31,32,33]. In response to growth factors, sphingosine kinase (SphK)-mediated production of S1P promotes cell proliferation in quiescent Swiss 3T3 fibroblasts via activation of dual signal transduction pathways, including the release of calcium from internal stores independent of Inositol triphosphate and activation of phospholipase D [34,35]. As opposed to S1P, ceramide, the backbone of all sphingolipids and precursor to S1P, acts as a growth arrest signal and promotes apoptosis in response to cytokines [36]. Thus, the dynamic balance between S1P and ceramide plays an important role in cell fate determination [37]. In addition, the acyl chain composition of phosphatidylinositols has been linked to p53 mutations in cancer cells [38]. Although altered lipid metabolism is intimately associated with cancer development, and cytokinetic failures, not much is known about the interrelationships between lipids, cytokinesis, and the development of cancer.

## 2. Lipid Shapes, Asymmetric Distribution, Lateral Organization, and Polarization during Cytokinesis

Induction of the highest plasma membrane curvature at the site of cytokinesis requires remodeling of protein and lipid composition. Biophysical and chemical properties of lipids play essential roles in mediating such high membrane curvature. Lipid head group chemical composition, acyl chain length, and saturation dictate membrane bending plasticity (Figure 2). Several lipids, including cholesterol [39,40], phosphatidylinositol phosphates (PIPs) [41,42], gangliosides GM1 [40] sphingomyelin (SM) [43], and phosphatidylethanolamine (PE) [44], have been shown to be enriched at the cytokinetic furrow (Figure 2). Further, lipidomic analysis of purified midbodies from cultured HeLa cells and neuroepithelial cells from cerebrospinal fluid showed accumulation of several sphingolipids, phosphatidylserine (PS), phosphatidic acid (PA), PE, ether PE, ether phosphatidylcholine (PC), and triacylglycerols (TAG) with unique acyl chain composition [11,12].

Lipids with small head groups relative to their acyl chains, such as PE, can have a conical shape that allows the head groups of PE to come close together to favor concave bending of the membrane leaflet (Figure 2). Conversely, lipids with larger head groups relative to their acyl chains, such as PIPs, can have an inverted conical shape that confers convex membrane bending property to the leaflet (Figure 2). Lipids that have similar head and tail cross sectional area are cylindrical in shape and therefore do not influence membrane curvature (ex. PC and PS) [45,46]. However, successful bilayer bending may require both concave and convex bending lipids to be present on the opposite leaflets. For instance, at the cytokinetic furrows in mammalian cells, the concave bending lipid PE was shown to be enriched at the outer leaflet of PM; in contrast, PI(4, 5)P_2,_ a convex bending lipid, localized to the inner leaflet (Figure 2). Besides lipid composition, the successful development of essential membrane curvature is also influenced by protein composition, such as the shape of transmembrane proteins, protein–protein crowding [47], and by proteins that insert their amphipathic/hydrophobic helix (ADP ribosylation factors (Arf), Bin/Amphiphysin/Rvs (BAR) domain, etc.,) into the membrane [48,49,50]. Finally, the cytoskeleton is important in generating the force required to achieve high membrane curvature [51,52]. BAR domain proteins are important regulators of membrane curvature; they act as connecting links between the membrane and dynamic actin. BAR domain proteins are crescent-shaped, contain positively charged residues at the concave side that allow preferential binding to anionic phospholipids [53]. The F-BAR domain protein CDC15 plays important roles in the assembly of the contractile ring in the fission yeast Schizosaccharomyces pombe [54]. CDC15-mediated contractile ring assembly involves a sequence of events including dephosphorylation of the hyperphosphorylated state at interphase, binding to formin CDC12, localization to the site of cytokinesis, recruitment of Myo1 (class II myosin heavy chain), and activation of the Arp2/3 complex to promote actin cable network formation [53,55,56,57,58,59,60,61,62,63,64,65].

It is well known that several lipids are asymmetrically distributed between the inner and outer leaflets of biological membranes [66,67]. In mammalian cells, amino phospholipids such as PE and PS and negatively charged phospholipids such as phosphatidylinositols (PIs) are localized in the inner leaflet, whereas choline containing phospholipids such as PC and SM and glycosphingolipids (GSLs) are enriched in the outer leaflet of the plasma membrane. In contrast to mammalian cell plasma membranes, amino phospholipids PS and PE in insect plasma membranes are symmetrically distributed between inner and outer leaflets [68]. However, transbilayer and lateral organization of phosphatidylinositol phosphates and sphingolipids is conserved in all eukaryotic cells. The acyl chain composition is also asymmetrically distributed between the two leaflets; for instance, it was shown that the cytoplasmic leaflet is two-fold more unsaturated than the exoplasmic leaflet [69]. In addition to transbilayer asymmetry, lipids and proteins in the PM are segregated in the lateral dimension, whose behavior is best explained by three mutually non-exclusive models, including the picket-fence, lipid raft, and protein island models. Please see Box 2 for more details on each of these models. In the following subsections, we will discuss the role of individual lipids in cytokinesis in detail.

Box 2Models of plasma membrane organizationOur understanding of plasma membrane structure continues to evolve as newer imaging methods, such as single molecule imaging, become available. As a result, significant modifications are added to the textbook fluid mosaic model of the plasma membrane. Currently, plasma membrane structure and function are best explained by three mesoscale organizing principles, including actin-based partitioning of membrane proteins and associated lipids (the picket-fence model), the formation of liquid-ordered domains due to molecular affinities between lipids and proteins (the lipid raft model) and existence of proteins as dynamic molecular complexes (the protein island model).**Picket fence model:** Single molecule tracking studies have shown that diffusion of molecules in the plasma membrane of living cells is 5–50-fold slower than that in the artificial membranes [70,71,72,73]. Based on the experimental evidence, it was proposed that the plasma membrane is divided into unspecific membrane compartments (40–300 nm) where molecules are temporarily confined. Such molecular confinement is dependent on the actin cytoskeleton, as depolymerization of actin increases the confinement zones. These observations were interpreted as transmembrane proteins with their immediate lipid environment (pickets) being anchored to an actin-based membrane skeleton (fence) located parallel to the plasma membrane in the cytosol. Molecules show unrestricted diffusion within the confinement zone but undergo hop-diffusion when they enter the neighboring zone. Since the borders of confinement zones are unspecific, they cover the entire plasma membranes without affecting protein distribution, and the lipid rafts can coexist, although their dynamics are likely affected by diffusion barriers [70,74].**Lipid raft model:** Lipid rafts are small dynamic nanoscale (2–20 nm) assemblies composed of cholesterol, sphingolipids, and proteins that can be stabilized to grow into mesoscale (2–300 nm) functional domains [70,74,75,76]. Lipid rafts are also known as liquid ordered domains (lo domains) or detergent resistant membranes (DRMs). According to this model, at any given time, about 35 percent of all membrane proteins are localized to lipid rafts, and the rest of the 65% can move freely according to Singer and Nicolson’s fluid mosaic model [77,78]. Lipid raft dynamics can also be influenced by actin cytoskeleton via transbilayer coupling involving long acyl chain interdigitation and in the cytoplasm, phosphatidylserine head group to cytoskeleton interactions via adaptor proteins [79,80,81,82,83]. An emerging concept called “active emulsions” suggests that lipid raft formation and growth into mesoscale domains require transbilayer coupling, cortical actomyosin contraction, and lateral molecular interactions [81,83]. Thus, the active emulsion concept connects the lipid raft with the cortical actin and cortical flows to mediate the formation of mesoscale nonequilibrium lateral membrane organization [81,83].**Protein Island model:** Based on the electron and super-resolution microscopy analyses, it was proposed that all the plasma membrane proteins are segregated according to their nature and function into distinct domains called protein islands. These domains can be classified as raft or non-raft depending on their interactions with the lipid molecules; additionally, transmembrane proteins can interact with the actin cytoskeleton for dynamic movement. Molecules undergo hop-diffusion when they translocate from one protein island to another via shared borders. The protein-free and low cholesterol membrane domains separate these islands [84,85].

## 3. Phosphatidylethanolamine (PE)

In mammalian cells, PE is normally localized to the inner leaflet of the PM; however, during cytokinesis, it gets redistributed to the outer leaflet of the cleavage furrow. Immobilization of cell surface PE by PE binding peptide or reduction of PE levels by genetic ablation of PE synthesis in CHO cells specifically blocked disassembly of the contractile ring after furrow ingression, leading to the formation of a long cytoplasmic bridge between daughter cells. This phenotype was rescued by the addition of PE or its precursor ethanolamine, suggesting an important role for transbilayer PE redistribution in the disassembly of the contractile ring [44]. Immobilization of PE at the outer leaflet of the plasma membrane led to PI(4,5)P_2_ over accumulation in the intercellular bridge, prevented actin disassembly, and inhibited abscission [41].

The acyl chain composition of PE is also important in cytokinesis. The ether-linked PE species with longer and more unsaturated fatty acids accumulated in the midbody lipidome of mouse neuroepithelial cells. In contrast, phosphatidylcholine species were composed of shorter and more saturated acyl chains [11]. In *Arabidopsis thaliana,* depletion of very long chain fatty acids (VLCF) by mutation in a microsomal elongase gene significantly reduced VLCFA-containing PE species and induced cytokinetic defects by affecting endomembrane dynamics [86]. Thus, PE redistribution and its acyl chain composition may play an important role in cytokinesis via regulation of PI(4,5)P_2_ levels [41].

## 4. Phosphatidylinositol Phosphates (PIPs)

Phosphatidylinositols (PI) are glycerophospholipids with a myo-inositol head group. PIs are initially synthesized on the membranes of the endoplasmic reticulum and subsequently transported to other cellular membranes. The cytosolically exposed hydrophilic myo-inositol head group on the target membranes gets modified by selective PI-kinases or PI-phosphatases to generate unique PIP domains in response to various cellular cues [87]. Phosphorylation of the myo-inositol head group generates seven forms of phosphatidylinositol phosphates (PIPs) including three monophosphates (PI3P, PI4P, and PI5P), three bisphosphates [PI(4,5)P_2_, PI(3,4)P_2_, and PI(3,5)P_2_] and one triphosphate, PI(3,4,5)P_3_ [88,89]. Importantly, specific PI functions are achieved through the recruitment of specific PI-binding proteins that are selective for one or a few PIs (e.g., Anillin’s PH domain binds to PI(4,5)P_2_ or FYVE-CENT binds to PI(3)P). PIPs are uniquely distributed in the subcellular membrane compartments. PI(4)P is localized to PM, endosomes, and the trans Golgi. PI(3)P is found in early endosomes, whereas PI(5)P is found on the PM, endosomes, nuclear envelope. PI(4,5)P_2_ localized to PM, recycling endosomes, and lysosomes; PI(3,4)P_2_ to PM and early endosomes; PI(3,5)P_2_ to late endosomes and lysosomes; PI(3,4,5)P_3_ is localized to the PM and endocytic compartments [90]. To date, several PIs have been implicated in cell division and cytokinesis, including, PI, PI3P, PI4P, PI(4,5)P_2_, PI(3,4)P_2_, and PI(3,4,5)P_3_ [3,4,10,88,89].

PI(4,5)P_2_ is the most extensively studied lipid in animal cell cytokinesis, owing to its obvious enrichment in cytokinetic furrows in various cell types and organisms, as well as its broad implications in lipid–lipid and lipid–protein interactions and signaling [21,23,41,42,43,91,92,93,94,95]. Disruption of PI(4,5)P_2_ levels is associated with cytokinetic defects in several model organisms and mammalian cells [40,96,97,98,99]. PI(4,5)P_2_ normally localizes uniformly to the inner leaflet of the plasma membrane in mammalian cells during prophase; however, during cytokinesis, PI(4,5)P_2_ is specifically enriched at the inner leaflet of the cleavage furrow [41,42]. However, PI(4,5)P_2_ accumulation at the furrow is not universal because its enrichment was not observed in *Drosophila* spermatocytes [94,99] and *Dictyostelium* [100], suggesting the existence of alternative mechanisms in different cell types or organisms. PI(4,5)P_2_ is required for adhesion of plasma membrane to the contractile ring [42]. In yeast, PI(4,5)P_2_ is required for localization of RhoA, a small GTPase required for assembly of the contractile ring [101,102]. Many cytoskeletal elements and regulatory proteins, including RhoA [43,102,103,104,105,106], Ect2 RhoGEF [103,105], Anillin [107], Syndapin [94], Septins [108,109,110,111], MgcRacGAP [112] and subunits of the vesicle-tethering complex Exocyst [113], bind to PI(4,5)P_2_. Furthermore, PI(4,5)P_2_ regulates cytokinesis via exocytosis, endocytosis, and endocytic recycling [6,114,115,116,117,118,119,120,121,122,123,124,125]. Recently, it was shown that a phosphatidylinositol-4-phosphate 3-kinase catalytic subunit type 2A (PI3KC2A) localizes to the midbody via binding to PI(4,5)P_2_ and γ-tubulin and produces PI(3,4)P_2_ from PI(4)P. The ESCRT II component VPS36 binds to PI(3,4)P_2_ and recruits, ESCRT III component CHMP4B in an ALIX independent pathway to mediate abscission in eye lens cells [10]. Besides PI(4,5)P_2_ production, its turnover and clearance also influence cytokinesis [41,92,99]. Several excellent reviews have focused on PIs and their role in cytokinesis and will not be discussed extensively here [3,4,88,89,126,127,128,129].

In mitotically dividing cells, PI(3) kinase and its product PI(3,4,5)P_3_ localized to polar regions; in contrast, PTEN, a PI(3,4,5)P_3_ phosphatase, localized to concentrates at the cleavage furrow [100]. Similarly, PI4P 5-kinase and its product PI(4,5)P_2_ accumulate at the furrow, thus effectively enriching PI(4,5)P_2_, and depleting PI(3,4,5)P_3_ at the furrow [42,130]. Interestingly, we have shown that a PI(3,4,5)P_3_ specific probe tGPH (GFP fused to pleckstrin homology domain of Grp1/Steppke under tubulin promoter) was enriched at the cleavage furrow in dividing *Drosophila* spermatocytes indicating PI(3,4,5)P_3_ localization to this region. However, tGPH enrichment at the furrow was observed only in intact cysts, i.e., spermatocytes encased by cyst cell membrane, and acute loss of cyst cell membranes and concomitant reduction in PI(3,4,5)P_3_ accumulation at the furrow did not prevent ingression of cleavage furrow during cytokinesis in spermatocytes [131]. The significance of such PI(3,4,5)P_3_ enrichment at the furrow is currently unknown and warrants future investigation. A study on plasma membrane expansion during cellularization in *Drosophila* embryos found that PI (3,4,5)P_3_ inhibited PI(4,5)P_2_-dependent actomyosin contractility, promoting actomyosin network disassembly [132]. The PI(3,4,5)P_3_ probe, Steppke, localized to actomyosin networks and reduced tissue tension by inhibiting the actomyosin activity at the adherens junction during dorsal closure in *Drosophila* embryos [133]. Thus, PI(4,5)P_2_ and PI(3,4,5)P_3_ are likely to play distinct and antagonistic roles in the regulation of actomyosin ring contractility during cytokinesis.

## 5. Sphingolipids

Sphingolipids are essential for cytokinesis, as disruption of sphingolipid biosynthesis either by genetic ablation or pharmacological methods induces cytokinetic defects [12,134,135,136]. It was shown that inhibition of sphingolipid synthesis reduces cell surface area by 45% and disrupts the actin cytoskeleton in Swiss 3T3 cells, and these effects are reversed by the addition of ganglioside GM3 [135]. Inactivation of glucosyl ceramide synthase (GCS) either by RNA interference (RNAi) or by 1-phenyl-2-palmitoyl-amino-3-morpholino-1-propanol (PPMP) caused failure of cleavage furrow ingression [134] due to accumulation of C16, C20, and C22 ceramides. Ceramides added exogenously to cultured cells caused cytokinetic defects. It was shown that inhibition of GCS results in mislocalization of cytoskeletal proteins, including actin and ERMs (ezrin, radixin, and moesin) that connect the PM with the actin cortex [134]. PPMP was also shown to induce cytokinetic defects in Giardia lamblia [136]. Long-chain sphingolipids, including dihydroceramide (C22 and C24), ceramide (C24), and hexosylceramides (C16 and C24) species, were found in the midbody lipidome of cultured HeLa cells, and RNAi knockdown of various sphingolipid metabolic enzymes, including biosynthetic and catabolic enzymes, resulted in cytokinetic defects [12]. Sphingolipids were shown to be essential for cytokinesis in *Trypanosoma brucei*. Depletion of serine palmitoyl transferase subunit 2 (SPT2) by RNAi or its inhibition with myriocin affected cell cycle progression but not vesicular trafficking or lipid raft formation [137]. Very long acyl chain (C24) glycosphingolipids (GSLs) such as glucosyl-ceramides were shown to be essential for cell plate formation during cytokinesis in plants [138].

Sphingolipids such as SM and gangliosides in combination with cholesterol were shown to form liquid ordered domains (L_o_ domains) on the outer leaflet of the PM [75]. Ganglioside GM1 and the cholesterol-rich domains were shown to be enriched in cleavage furrows of dividing sea urchin eggs [40]. Accumulation of GM1 and the cholesterol-rich domain at the furrow is dependent on the onset of anaphase, myosin light chain phosphorylation, actin, and microtubules. Saturated fatty acids but not unsaturated fatty acid-containing lipids were found to accumulate at the furrow. Membrane-associated proteins such as Src (non-receptor tyrosine kinase) and PLC _γ_ (phospholipase C) are enriched in L_o_ domains, and their activation via tyrosine phosphorylation is essential for cytokinesis [40]. PLC _γ_ activation leading to Ca^2+^ release is essential for cytokinesis [98]. Inositol 1,4,5-triphosphate (IP3) receptor-mediated calcium release is required for furrowing in spermatocytes and sea urchin eggs [99,139].

Sphingomyelin (SM), a major sphingolipid, is normally present uniformly on the outer leaflet of PM, and, during cytokinesis, it specifically gets enriched at the cleavage furrow. Interestingly, SM at the outer leaflet regulates PI(4,5)P_2_ accumulation in the inner leaflet of the furrow [43]. Depletion of SM at the outer leaflet using SMase treatment specifically disrupts PI(4,5)P_2_ but not cholesterol accumulation [43]. Cleavage furrow-specific enrichment of SM is dependent on cholesterol but not on PI(4,5)P_2_. Although it is unclear how SM regulates PI(4,5)P_2_ enrichment at the furrow, it has been proposed that SM could directly or indirectly interact with PI(4,5)P_2_ to limit its diffusion as well as locally activate PI(4)P 5-kinase for confined PI(4,5)P_2_ synthesis [43]. Proteins embedded in the cytoplasmic leaflet of the cleavage furrow indeed show slower diffusional rates than the outer leaflet, and this process is regulated by septins in mammalian cells [140,141,142]. Thus, sphingolipids regulate cytokinesis via the formation of L_o_ domains, transbilayer coupling, signal transduction, and cytoskeletal reorganization.

## 6. Cholesterol

Cholesterol was shown to accumulate at the cleavage furrow in sea urchin eggs [40], more specifically at the outer leaflet of the cleavage furrow, whereas it remained uniformly distributed in the inner leaflet [43]. Sterol rich membrane domains have been identified in growing tip and septum of *S. pombe*, and their integrity is essential for proper cytokinesis [143,144]. Optimal sterol levels are important for normal cytokinesis; in fission yeast, it was shown that increasing ergosterol levels delays formin cdc12-dependent assembly of F-actin and disrupts division plane positioning [144,145]. In human tissue culture cells, cholesterol depletion results in polyploidy [39]. L_o_ domain-associated proteins flotillins were shown to form large, stable domains in the plasma membrane and accumulate at a higher density in the cleavage furrow of hematopoietic cells [146]. Cholesterol was shown to mediate local endocytosis in the ICB via the formation of midbody tubules, and that depletion of cholesterol prevents the formation of such midbody tubules and induces cytokinetic defects [147]. Caveolae are cup shaped 50 to 100 nm plasma membrane invaginations enriched in cholesterol and sphingolipids [148,149,150,151]. Caveolae were shown to be enriched at the cleavage furrow, and intercellular bridge, and midbody in mammalian cells and early zebrafish embryos [152,153,154]. Functionally, caveolae at the intercellular bridge were shown to buffer membrane tension and limit contractibility to promote ESCRT-III assembly and cytokinetic abscission [152].

## 7. Triacylglycerols (TAG)

Budding yeast quadruple mutants are1Δ, are2Δ, dga1Δ, and Iro1Δ, that cannot synthesize triacylglycerols show abnormal cytokinesis and septation defects [155]. TAGs with short chain acyl chains (16:1/12:0/18:1) have been shown to be specifically enriched in purified midbodies of HeLa cells and neuroepithelial cells [11,12]. RNAi knockdown of TAG metabolic enzyme DGAT2 significantly increased cytokinetic defects in HeLa cells [12]. Although TAG accumulation at the midbodies is intriguing, its specific mechanism of action remains unknown [8].

## 8. Phosphatidic Acid

Phosphatidic acid (PA) is yet another phospholipid enriched in purified midbodies from HeLa cells, and RNAi knockdown of its metabolic enzyme ABHD5 resulted in cytokinetic defects [12]. However, the mechanism by which PA regulates cytokinesis remains an open question. Phosphatidic acid is a conical shaped lipid that is localized to the cytosolic leaflet of the PM and imparts concave membrane bending properties. Phosphatidic acids are important signaling molecules, involved in vesicular trafficking, and are also precursors for the synthesis of most of the glycerophospholipids, such as phosphatidylserine and phosphatidylinositols [156,157,158]. PA has been shown to stimulate PI(4)P 5-kinase via Arf6 to promote PI(4,5)P_2_ synthesis [159,160,161]. PA can also recruit sphingosine kinase to the plasma membrane, where it catalyzes the synthesis of sphingosine-1-phosphate, an important signaling molecule [157]. Inhibition of sphingosine kinase 1 compromises PKC activity and cytokinesis [162]. Phosphorylation of nonconventional PKC is required for completion of cytokinesis, and inhibition of it results in sustained RhoA activation and delayed contractile ring disassembly [163,164]. The *Drosophila* sphingosine-1-phosphate lyase gene (Sply) and sphingosine kinase 2 were shown to be important for normal muscle development and reproductive organ function [165,166], but no direct role in cytokinesis has been demonstrated.

## 9. Phosphatidylserine (PS)

PS is synthesized in the endoplasmic reticulum and trafficked via the Golgi to the plasma membrane, where it is mostly localized to the inner leaflet. In animal cells, PS can be flipped from the inner leaflet to the outer leaflet, where it acts as ‘eat me’ signal that promotes phagocytosis to engulf the cells. Apoptotic cells activate scramblases that quickly expose PS to the outer leaflet [167]. Relatively little is known about the role of PS in cytokinesis, although its specific enrichment was observed in the purified midbody lipidome of HeLa cells, particularly unsaturated long-chain PS [12]. In fission yeast, genetic ablation, or overexpression of Phosphatidylserine synthase (pps1), induced cell morphology and cytokinetic defects, suggesting a dose-dependent role for PS in cytokinesis [168]. Recently, it was shown that PS with very long-chain fatty acids are required for cytokinesis in Arabidopsis roots, and it was suggested that PS may help mediate cargo vesicle trafficking during cell plate formation [169]. Anionic phospholipids like PS and PA likely mediate important functions in cytokinesis by binding to various peripheral membrane proteins that have lipid binding domains such as BAR domain proteins, Annexin repeats, and C2 domain containing proteins (Table 1). Future studies are required to demonstrate the direct involvement of PS in animal cell cytokinesis.

## 10. Sphingolipid Acyl Chain Composition in Male Meiotic Cytokinesis

Cytokinesis in somatic cells results in the complete physical separation of cytoplasm between the daughter cells. In contrast, cytokinesis in spermatocytes is incomplete, where all the sister cells are interconnected via cytoplasmic bridges [259]. The presence of sphingolipids (ceramides, sphingomyelin, and glycosphingolipids) with very long chain polyunsaturated fatty acids (VLC-PUFA) is one of the characteristic features of mammalian spermatozoa/testes [260,261,262,263,264,265]. In general, sphingolipids are required for cytokinesis in both somatic cells and male meiotic cells; however, VLC-PUFAs containing sphingolipids are specifically associated with male meiotic cells [263]. Glycosphingolipid deficiency leads to male meiotic cytokinesis defects [266] and is required for intercellular bridge stability [267]. Ceramide synthase 3 (CerS3) was discovered to be required for the synthesis of sphingolipids containing very long chain polyunsaturated fatty acids [266,267]. A *Drosophila* member of the very long chain fatty acid synthase (Elovl) gene, Bond, was shown to be important for cytokinesis in spermatocytes [119]. Bond mutants show reduced cleavage furrow ingression with frequent contractile ring detachment from the cortex, constriction, and collapse of the contractile ring to one side of the cell, leading to cleavage furrow regression [268]. Lipids containing very long chain fatty acids have been proposed to promote membrane deformation and stable contractile ring and plasma membrane interaction during cytokinesis [268]. *Drosophila* desaturase gene family members exhibit a sex-specific expression pattern [269]. Sphingolipid delta-4 desaturase (Des1/ifc1) catalyzes the final step of de novo ceramide synthesis by introducing a double bond in the sphingoid base and is essential for spermatogenesis but not for oogenesis [270,271]. P-element insertion mutants of des-1 were shown to be defective in central spindle assembly during male meiotic cytokinesis. During anaphase and telophase, DES-1 localizes to mitochondria along the spindle apparatus. Before the biochemical identification of this protein as a sphingolipid desaturase enzyme, it was proposed that DES-1 could act as an anchoring mechanism where it links membrane-bound cellular compartments to the cytoskeletal components [270].

The small regulatory subunit of serine palmitoyltransferase (ssSPT), Ghiberti (also known as frodo), which is part of the Serine Palmitoyltransferase (SPT) enzyme complex, has been shown to be required for male meiotic cytokinesis [119,272]. *ghi* mutants display normal actomyosin ring assembly, but the ring fails to constrict to completion, leading to furrow regression and cytokinetic defects [119]. Loss of *Ghi* affects the d16/d14 (sphingoid base acyl chain length) ratio, with a net shift towards increased accumulation of sphingolipids with a d16 sphingoid base, suggesting a crucial role for sphingoid base acyl chain length in male meiotic cytokinesis [272]. Further, *Drosophila* males had more unsaturated fatty acids in their sphingolipids compared to females [272]. However, the molecular mechanisms by which these specific sphingolipids regulate cytokinesis were unclear until recently. Recent studies, including ours, showed that de novo biosynthesis of ceramide posphoethanolamine (CPE), a sphingomyelin analogue in *Drosophila,* is essential for male fertility [273,274]. We have shown that testis specific CPEs show increased unsaturation both in the sphingoid base and fatty acid acyl chains (e.g., d14:1/C24:1, and d14:2/C24:0). The ethanolamine head group of CPE is also essential for cytokinesis, as expression of human sphingomyelin synthase that produced SM in *Drosophila* spermatocytes did not rescue cytokinetic defects [131]. Live cell imaging analysis using the mushroom-derived CPE-binding protein pleurotolysin A2 (PlyA2) showed that CPE is endocytosed from the plasma membrane and delivered to the cleavage furrow via the endocytic pathway, and this CPE trafficking is essential for male meiotic cytokinesis (Figure 3) [131].

Why do male germ cells have high amounts of unsaturated fatty acid-containing lipids, and why are they important in male meiotic cytokinesis? Although exact mechanisms are currently unknown, the answer could lie in their widely known roles in signaling and modulation of membrane biophysical properties. Unsaturated fatty acid-containing lipids introduce more disorder in the membranes; thereby, they make membranes more fluid, reduce membrane thickness, increase lipid packing defects, reduce membrane rigidity/make membrane more flexible, affect membrane protein conformations, and increase sensitivity to oxidation (Figure 2) [275,276]. Sphingolipids with unsaturated fatty acids were found to be excluded from liquid-ordered domains/lipid rafts in rat spermatocyte membranes [277]. Sphingolipids with unsaturated fatty acids may thus increase membrane flexibility or curvature and or lipid–protein interactions during male meiotic cytokinesis and this requirement may be different in the case of somatic cell cytokinesis. For instance, membrane bending is not only important at the ingressing cleavage furrow but also in other cellular processes such as endocytosis and the maturation of multivesicular endosomes. It was shown that when polyunsaturated fatty acids (PUFA) containing glycerophospholipids (GPL) were present at the convex leaflet of a bent membrane, they reduced the rigidity [278] and facilitated endocytosis [279]. Indeed, it was shown that PUFA-GPLs are asymmetrically distributed in mammalian cell plasma membranes and the endocytic system, where the cytoplasmic leaflet is two-fold more unsaturated than the exoplasmic leaflet and that the outer PM leaflet is more packed and rigid [69]. However, in contrast to glycerophospholipids, sphingolipids such as sphingomyelin or glycosphingolipids are normally localized to the outer leaflet of the plasma membrane [69]. Thus, unsaturated fatty acids in the outer leaflet localized sphingolipid could have a unique convex membrane bending property, as in the case of multivesicular endosome maturation. Indeed, the sphingolipid ceramide was shown to be required for the formation of intraluminal vesicles in multivesicular endosomes via an ESCRT (endosomal sorting complex required for transport) independent mechanism [280] and the absence of DES1 significantly reduced the MVB intraluminal vesicle density [281]. In concurrence, our recent study showed that CPE is enriched in intraluminal vesicles of multivesicular endosomes, which in turn localize to cleavage furrow [131]. Together, these studies suggest that sphingolipids with unsaturated fatty acids, as opposed to saturated fatty acids, are excluded from liquid-ordered domains, reduce membrane rigidity, and facilitate endocytic membrane traffic during male meiosis cytokinesis.

## 11. Membrane Trafficking and Lipid Polarization at the Cytokinetic Furrow

A typical animal cell requires a total surface area increase of about 26% for successful division, and meiotic cells require about 60% due to two rounds of rapid successive divisions [192,282,283]. Such a dramatic increase in cell surface area would require multiple mechanisms of membrane trafficking and lipid remodulation, including (I) expansion of plasma membrane reserves present in the form of microvilli or other membrane projections (II) secretion of newly synthesized membranes via forward trafficking (III) membrane recycling via the endocytic pathway, and (IV) lateral movement of lipid domains and local synthesis [114,116,284]. In the following sections, we will discuss more about each of these mechanisms, with a specific emphasis on phosphoinositides and sphingolipids.

## 12. Regulation of Plasma Membrane Area during Mitosis

Most of the cultured animal cells entering mitosis dramatically change their shape from being flat at the interphase to round/spherical at the metaphase [284,285,286]. In addition to the shape, cells also significantly increase their volume and hydrostatic pressure from prophase to prometaphase [285]. By altering size and shape, cells provide a suitable environment for spindle formation and ensure proper segregation of chromosomes and organelles to daughter cells [285,286,287]. Mitotic cell rounding accompanies a significant reduction in cell surface area [284]. To accommodate the apparent change in cell area, the plasma membrane folds into microvilli, blebs, and ruffles (Figure 4) [288,289,290]. In addition, membranes are redistributed into endocytic vesicles that could potentially fuse with the PM at the later stages of cell division, including anaphase and telophase [291]. Using live cell imaging techniques, Boucrot, E. et al. discovered that the cell surface area decreases significantly during metaphase [284]. The decrease in cell surface area from prophase to metaphase coincides with increased endocytosis and reduced recycling. Starting at anaphase, membrane recycling resumes, resulting in the recovery of the cell surface. During this time, recovery of total cell surface occurs even in the absence of a functional Golgi [284]. We recently discovered that during male meiosis cytokinesis, CPE lipids in the plasma membrane are endocytosed and delivered to the cleavage furrow [131]. Together, these observations suggest that expansion of plasma membrane projections and endocytic recycling contribute to an increase in cell surface area during anaphase and cytokinesis. However, changes in cell shape and size have not been extensively studied in the context of crowded tissue environments and therefore warrant future investigations [285].

## 13. Forward Trafficking

Vesicle transport from the ER to the Golgi and then to the cleavage furrow was shown to be essential for cytokinesis in multiple animal model organisms, including Xenopus [292], *Caenorhabditis elegans* embryos [293], *Drosophila* embryos [294,295], and mammalian cells [124], and to the cell plate in plants [296,297]. Several lines of evidence from the studies conducted on *Drosophila* spermatocytes undergoing meiotic divisions have demonstrated crucial roles for Golgi-derived vesicular trafficking in cytokinesis [130,192,199,298,299,300,301,302,303,304,305]. They include essential functions of the small GTPase Rab1, the vesicular coat component COP-I, the tethering machineries (conserved oligomeric complex subunits Cog5 (FWS) and Cog7, and the transport particle (TRAPPII), and the endoplasmic reticulum-golgi trafficking regulator Zw10 of the Rod-Zwiltch-Zw10 (RZZ) complex and exocyst components Exo8 and Exo84, syntaxin 5, and PI(4)P binding protein Golgi phosphoprotein 3 (GOLPH3) [192,199,221,298,300,303,306,307,308]. Given that PI is a lipid that cannot diffuse freely across the cytosol from the ER, it is likely that the supply of PI from the ER is mediated by two processes, namely vesicular transport and via PI transfer proteins (PITPs) [309]. Importantly, phosphatidylinositol transfer protein (PITP) (Gio/Vib) and the PI4-kinase (PI4K), four-wheel drive (FWD), were shown to be important for cytokinesis in spermatocytes [130,301,310]. PITP is localized to the ER, cleavage furrow, and spindle envelope and is required for the fusion of Golgi-derived vesicles to the cleavage furrow [310]. PITP mutants show normal central spindle assembly, but furrow ingression is delayed, and they display an apparent loss of contact between the contractile ring and plasma membrane, leading to the regression of the furrow [301,310]. FWD localizes to the Golgi, and its activity is required for accumulation of PI4P on Golgi membranes as well as the midzone during the late steps of cytokinesis. FWD also physically interacts with Rab11 and helps in its recruitment to Golgi membranes, where it becomes associated with organelles containing PI4P. Thus, PI4Kβ has both catalytic and non-catalytic functions in promoting Rab11 localization during cytokinesis [282]. Similarly, in mammalian cells, PI4Kβ binds and recruits Rab11 to the Golgi where Rab11 plays a role in post-Golgi secretory trafficking [311,312]. Wild-type activity of PI4K is required for the appearance of tyrosine phosphorylation epitopes at the furrow, and normal organization of actin filaments in the contractile ring, and formation of intercellular bridge [130]. PI4P is a substrate for the synthesis of PI(4,5)P_2_, a major phosphoinositide, by PI4P 5-kinase, which is found in the cytokinetic furrow and intercellular bridges and whose enzymatic activity is required for cytokinesis [41,95]. PITP was shown to regulate asymmetric division in *Drosophila* neuroblasts by promoting the synthesis of a plasma membrane-specific pool of PI4P by PI4KIIIα. Further, PI4P in turn was shown to bind to non-muscle myosin II regulatory light chain (Sqh) in vitro, suggesting a mechanism for anchoring of myosin to the cell cortex in neuroblasts [313]. A study in *S. pombe* revealed that scaffolding protein efr3-mediated PI4-kinase PM attachment is required for central positioning of contractile rings, without which contractile rings slide to one end in a myosin-V-dependent manner [255,256]. GOLPH3 is an effector of PI4P that was shown to accumulate at the Golgi and cleavage furrow. Binding of GOLPH3 to PI4P is essential for localization of PI4P-enriched and Rab11-positive vesicles at the cleavage furrow [199]. GOLPH3 interacts with non-muscle myosin II regulatory light chain (Sqh) and Centralspindlin complex subunit Pavarotti (a kinesin-like protein) to regulate furrow ingression [306].

Relatively little is known about the role of other lipids in forward trafficking and cytokinesis. Very long chain sphingolipids have been shown in plant cells to be required for the fusion of Golgi-derived vesicles to the cell plate during cytokinesis [138]. Sphingolipids with very long chain fatty acids (C24, C26) at the trans Golgi were shown to regulate PI4P homeostasis by recruitment of phosphoinositide specific phospholipase C (PI-PLC) and depletion of PI4P at the trans Golgi during sorting of auxin efflux carrier PIN2 in plants [314]. Collectively, these studies suggest a crucial role for lipid trafficking, especially PI, PI(4)P and very long chain sphingolipids, via directed secretion in cytokinesis.

## 14. Endocytosis and Recycling

It was once believed that endocytosis was inactive during cell division. However, subsequent studies demonstrated that endocytosis is active and has an important role during cytokinesis [284,315,316,317,318,319,320]. At the onset of mitosis, during prophase, and metaphase mother cells round up, leading to a substantial reduction in cell surface area [2,284]. Live cell imaging in HeLa/BSC1 cells showed that continued endocytosis and a considerable reduction in membrane recycling back to the PM are responsible for this cell rounding. Reactivation of endosome fusion with the PM during anaphase and telophase rapidly recovers the cell surface area. Late endosomes and recycling endosomes both participate in membrane redeposition. This modulation of endosomal recycling is essential for cell division, as inhibition of endocytosis or recycling impairs cytokinesis [284]. Similar endosome fusion with the PM during cytokinesis has been found in other cell lines and organisms [294,319,321,322].

Pioneering studies conducted on *Drosophila* cellularization in embryos revealed an important role for Rab11-positive recycling endosomes in cytokinesis [323,324] and were also observed subsequently in mammalian cells undergoing mitosis [232]. Several studies examining cytokinesis identified Rab11 family interacting proteins 3 (FIP3) and 4 (FIP4) and Arf6 as Rab11 interacting proteins [231,232,325]. Rab11-FIP3 endosomes play a critical role in inducing secondary ingression in the late steps of cytokinesis [213,232,326,327]. It was shown that FIP3-endosomes are required for the delivery of protein cargoes SCAMP2/3 and p50RhoGAP to the intercellular bridge. p50RhoGAP inactivates Rac/Rho GTPases to promote cortical actin depolymerization in the intercellular bridge before secondary ingression during abscission [213]. siRNA screens for Rab GTPases involved in cytokinesis identified Rab35 in addition to Rab11 as an important regulator of cytokinesis both in *Drosophila* and mammalian cells [92]. Rab35 has been shown to reduce PI(4,5)P_2_ levels at the intercellular bridge via direct interaction with PI(4,5)P_2_ phosphatase (OCRL) to promote localized cortical actin remodeling required for intercellular bridge stability and abscission [23]. The formation of an actin-free zone in the intercellular bridge is thought to be important for vesicle fusion and/or the promotion of increased membrane dynamics [6,7]. Sagona AP, et al. identified a third class of endosomes that are enriched in phosphatidylinositol-3-phosphate and accumulate at the intercellular bridge [258]. These endosomes contain centrosomal protein FYVE-CENT and its binding partner TTC19, which in turn binds to CHMP4B, a subunit of the endosomal sorting complex required for transport (ESCRT) III, which is involved in the final abscission step of cytokinesis [258]. It was suggested that endosomal transport and fusion with the PM during cytokinesis have two major roles: addition of new membranes or specific lipids to the furrow/intercellular bridge, and fast delivery of regulatory proteins involved in the reorganization of the cytoskeleton and PM that is essential for successful abscission [7]. Besides membrane addition at the cleavage furrow, new membrane addition at the cell poles was also reported during anaphase and was dependent on actin and astral microtubules. The lipid composition of the polar region was shown to be distinct and was devoid of GM1 [328].

Recently, we have shown that the sphingolipid CPE is enriched in Rab7 (which marks late endosomes) and Rab11-positive endosomes that are specifically localized to the cleavage furrow during male meiosis cytokinesis. Further, localization of CPE-enriched endosomes to the cleavage furrow is dependent on the wild-type function of Rab11. The absence of CPE or interfering with the functions of Rab7, Rab11, or Rab35 significantly increases cytokinetic defects, suggesting that specific lipid traffic via the endocytic pathway is essential for cytokinesis. However, unlike Rab7 and Rab11 endosomes, Rab35 did not localize to the endosomal compartment in *Drosophila* spermatocytes; instead, it localized to mitochondrial membranes [131], suggesting that Rab35 may function differently in spermatocytes.

Membrane trafficking via the late endosome was also shown to be important for cytokinesis in mammalian cells [284]. The presence of a large number of multivesicular endosomes (MVEs) (named lytic endosomes in the study) in close proximity to the secondary ingression site was easily observable in the original EM images of the intercellular bridge [326], suggesting a conserved role of MVEs in cytokinesis. In HeLa cells, clusters of lysosomes were shown to localize to the intercellular bridge, and these clusters were regulated by calcium-binding protein-7 and PI4PIIIβ kinase [329]. Recently, these lysosome clusters were shown to fuse with the cleavage furrow and release their intraluminal contents outside of the cell as exosomes, and this process is important for cytokinesis (Figure 4) [330]. However, in *Drosophila* spermatocytes, fusion of MVEs with the cleavage furrow was not observed; instead, it was shown that MVEs release their intraluminal vesicles in the cytosol in the vicinity of the growing furrow, suggesting membrane delivery to the furrow membranes via small pockets (Figure 4) [131]. Together, these studies suggest that Rab11-positive recycling endosomes and late endosome mediated membrane trafficking is conserved between mammalian cells and *Drosophila* spermatocytes. However, the specific mechanism by which Rab35 mediates membrane trafficking during male meiotic cytokinesis in *Drosophila* spermatocytes remains to be investigated in the future.

## 15. Local Synthesis and Cortical Flow

PIPs are low-abundance, cytoplasmic leaflet-exposed phospholipids, yet they regulate several cellular processes [90,126]. The synthesis of different PIP species is tightly regulated both spatially and temporally by the action of kinases, phosphatases, and phospholipases that localize to different subcellular compartments [90,331]. Emoto et al. demonstrated that PI(4,5)P_2_ and its biosynthetic enzyme PI(4)P 5-kinase are found in the cleavage furrow and that overexpression of kinase-deficient PIP5K causes cytokinesis defects in CHO cells [41]. The PI(4)P 5-kinase homologue, Its3, was found to concentrate on the septum of dividing cells in fission yeast *S. pombe*, and mutants of Its3 delayed cytokinesis [95]. In *Drosophila* cells, PI(4)P 5-kinase, Skittles, and PTEN, a PI(3,4,5)P_3_ phosphatase, both produce PI(4,5)P_2_ and localize to the cleavage furrow [93]. The enrichment of PI(4,5)P_2_ in furrow membranes and the localization of PI4(P) 5-kinase suggest that PI(4,5)P_2_ local synthesis is important during cytokinesis [40,42,92,144]. However, relatively little is known about how PI(4)P 5-kinase localization to the cleavage furrow is regulated. PI(4,5)P_2_ has been linked to detergent-resistant membranes in several studies [332,333,334]. It was shown that during interphase, PI(4)P 5-kinase localized to the inner leaflet immediately below the sphingomyelin rich lipid domains in the outer leaflet of the PM [43]. During cytokinesis, sphingomyelin-rich lipid domains accumulate at the cleavage furrow, which in turn coincides with the localization of PI(4)P 5-kinase to the cleavage furrow. When sphingomyelin-rich domains in the outer leaflet are depleted, PI(4,5)P_2_ accumulation in the inner leaflet is disrupted [43]. Microvilli formation in epithelial cells was shown to depend on clustering of sphingomyelin that co-clusters the transmembrane protein podocalyxin-1, which in turn recruits ezrin/radixin/moesin (ERM) binding phosphoprotein-50 (EBP-50) and PI(4)P 5-kinase [335]. The addition of lysogalactosylceramide/galactosylsphingosine (psychosine) to Namalwa cells induced cytokinetic defects via inhibition of SM-rich lipid domain formation and suppression of PI(4, 5)P_2_ production [336]. It is unclear how these interleaflet-coupled domains rich in sphingomyelin, PI(4,5)P_2_, and PI(4)P 5-kinase concentrate at the cleavage furrow. Anchored picket fence model, combined with a lipid raft and cortical flow mechanism involving the actin cytoskeleton, may explain this phenomenon, given that sphingolipids are not synthesized locally at the cleavage furrow (Figure 4) (please see Box 2 for more details on current plasma membrane models). A thin and dynamic network of filaments just under the plasma membrane is known as the cell cortex. Flow of cytoplasm in the cell cortex causes mechanical compression of filaments in the middle and aids in the formation of a contractile ring during cytokinesis [337,338]. During cytokinesis, cortical flow is directed from low tension to high tension, i.e., from the poles to the equatorial region, and this cortical flow is dependent on non-muscle myosin II activity [339,340,341]. Several studies have shown that sphingolipid-rich domains are regulated by cortical actin [79,342,343]. Lateral organization of lipids into nanoscale cholesterol-dependent domains requires long-chain saturated fatty acid-containing lipids in one of the two leaflets. Immobilization of long-chain saturated fatty acid-containing lipids in one leaflet affects the corresponding lipids in the opposite leaflet. Immobilization of inner leaflet lipids occurs upon association with the cortical actin cytoskeleton, leading to transbilayer coupling and the formation of nanoscale cholesterol-dependent lipid clusters resembling lipid rafts [81]. The acyl chains of the outer leaflet sphingolipid GM1 interdigitate with the acyl chains of the anionic phospholipid PS in the inner leaflet, and this interaction is enhanced by cholesterol [79]. Further interaction of the cortical cytoskeleton with the anionic phospholipid PS in the inner leaflet is required for the organization of sphingolipid rich domains in the outer leaflet [79]. When transbilayer interactions are coupled with active contractile flows (via interaction of inner leaflet lipid head groups and transmembrane proteins with the dynamic actomyosin), they create nanoscale clusters which in turn facilitate lateral lipid–lipid interactions, leading to the formation of active emulsions and mesoscale liquid ordered domains [80,83,344,345,346,347]. In addition to acyl chain saturation, lipid headgroup chemical and geometric features (such as lipid shapes) may influence lipid–lipid and lipid–protein interactions [69,348,349], and thus likely have an impact on mesoscale organization. Since the cytokinetic furrow is intimately connected with the juxtamembrane contractile cortex, it is likely that lipid organization and polarization in this area arises from a combination of the picket-fence model, the active emulsion model of lipid rafts, and cortical flows (Box 2).

In addition to the cortical cytoskeleton directing the movement of specific lipid domains and associated proteins to the cleavage furrow, anionic phospholipids in the inner leaflet of these lipid domains, such as PI(4,5)P_2_, may also direct cortical cytoskeleton organization via activation of specific signaling cascades. Consistent with this hypothesis, it was shown that cytoskeletal elements including microfilaments, myosin II, and microtubules are required for the formation of a cholesterol- and GM1-rich membrane band at the cleavage furrow in sea urchin eggs [40,350]. Additionally, these lipid domains contained signaling proteins Src and PLCγ, which are tyrosine phosphorylated during cytokinesis [40]. A systematic RNAi screen identified several GPCRs and their effectors involved in cytokinesis [200]. PI 4-kinase, which produces a precursor for PI(4,5)P_2_ biosynthesis, on the other hand, is required for tyrosine phosphorylation and normal actin filament organization in the contractile ring [130]. The cortical association of Moesin, an actin/membrane linker protein, is dependent on local production of PI(4,5)P_2_ in *Drosophila* [93]. Psychosine treatment of the myelomonocyte cell line U937 altered actin filament organization and caused cytokinetic defects [351]. Psychosine-mediated induction of cytokinetic defects also strictly depended on the G-protein-coupled receptor TDAG8 (T-cell death-associated gene 8) [17]. Thus, interplay between lipid domains, the cytoskeleton, and signaling proteins is crucial for the successful completion of cytokinesis.

## 16. Conclusions and Perspectives

Eukaryotic cells invest a significant amount of genetic information and energy to produce and degrade thousands of lipid species, whose complexity and biological roles are only beginning to be explored. Lipid composition dramatically changes between each cell type and within the same cell type depending on the different cellular states in which they are examined, for instance when they are dividing or differentiating [67,352]. In fact, cellular diversity is dependent on lipid diversity, which actively remodulates metabolic programs by regulating signaling receptors [352]. Although we now have a broad idea about the major types of lipid species associated with cell division and cytokinesis, we know relatively little about how these lipids regulate cytokinesis, especially lipids other than phosphoinositides. Even within phosphoinositides, we know little about their fatty acid acyl chain composition and how they affect cytokinesis. The lack of reliable probes to detect dynamic changes in lipid composition is one of the major limitations in studying these lipids. There are almost no in vivo tools available to study dynamic changes in acyl chain composition.

Midbody lipidomes revealed a number of novel lipids, including ceramides, ether phospholipids, phosphatidic acid, and triacylglycerol; however, their roles in cytokinesis remain unknown [12]. Sphingolipid-specific probes developed from pore-forming toxins such as Lysenin (binds to sphingomyelin), Pleurotolysin A2 (binds to CPE) and bacterial toxins such as Cholera toxin, Shiga toxins (binds to glycosphingolipids) have allowed analysis of dynamic changes of these lipids in the exoplasmic leaflet of cultured cells [352,353,354]. However, applications of these toxins are often limited to in vitro experiments. Hence, future studies focused on developing probes that could lead to better detection of sphingolipids in vivo would advance our understanding of the role of these enigmatic lipids in cytokinesis.

Long chain PUFAs as components of phospholipids and sphingolipids have been detected in the spermatozoa of most mammals, including humans and birds. Unsaturated fatty acid-containing lipids provide the spermatozoan plasma membrane with the fluidity and flexibility that it needs to participate in membrane fusion events during fertilization, although they make the plasma membrane more susceptible to oxidation [355]. Studies conducted in mice and *Drosophila* have shown that very long chain unsaturated fatty acid containing sphingolipids are essential for male meiotic cytokinesis [267,356]. Our recent study suggested the importance of endocytic trafficking of unsaturated fatty acid containing CPE via multivesicular endosomes and release of their intraluminal vesicles in the vicinity of furrow membranes [131]. However, several questions remain unanswered, including how ILVs released by MVBs get inserted at the cytokinetic furrow; do these endosomes have roles other than membrane addition at the furrow; do they deliver specific cargoes or mediate signaling; and finally, do similar mechanisms exist in mammalian male germ cells. Although the docking of lysosomes at the cytokinetic furrow of mammalian cancer cells resembles that of MVBs docking in *Drosophila* male meiotic cells, they fundamentally differ in the release of ILVs. Lysosomes release their ILVs outside the cell as exosomes when they fuse; in contrast, in our study, we find that MVBs release their ILVs in the cytosol proximal to ingressing membranes at the furrow [131,329,330]. These differences could be partly explained by the fact that male meiotic cells are unique in their lipid composition and polarization relative to somatic cells [267,356]. However, future studies are needed to shed light on mechanisms and their significance in cytokinesis of these cell types.

There is overwhelming evidence showing polarization of specific lipids at the cytokinetic furrow, and intercellular bridges [3,8]. However, we know very little about how lipids in the plasma membrane that are distributed uniformly in the prophase become concentrated and remain at the cytokinetic furrow during early to late telophase. Further, at the end of cytokinesis, the mechanisms by which these lipids achieve uniform distribution remain unknown, particularly when considering sphingolipids. Mechanisms involving transbilayer lipid domain coupling and their interactions with the cytoskeleton and lipid kinases is another area that needs future investigation. Furthermore, the role of the cortical flow model in lateral movement of lipid domains to the cleavage furrow via interaction with the cytoskeleton must be investigated. Especially due to a recent study conducted in *C. elegans* embryos warranted that, the cortical flows play only a minor role in actomyosin ring assembly and cytokinesis [357]. Several studies have now highlighted the importance of directed membrane secretion via the Golgi and membrane recycling via the endocytic pathway in the early and late steps of cytokinesis [3]. However, we lack much information on the specific cargos they carry and their role in cytokinesis. Future studies must therefore focus more on the identification of specific cargoes and their role in cytokinesis.

Lastly, sphingolipids and cholesterol-rich domains are known to co-cluster several transmembrane receptor proteins that mediate signaling [358]. Enrichment of these lipid domains at the cytokinetic furrow clearly suggests involvement of active signaling at the cytokinetic furrow and intercellular bridges. Although a few studies have found increased accumulation of tyrosine-phosphorylated proteins such as Src and PLCγ, their exact role in cytokinesis remains unknown [40]. Similarly, few GPCRs have been implicated in cytokinesis; however, mechanistic insights are still lacking [200]. Exploring the role of various signaling cascades in cytokinesis could refine mechanisms in cytokinesis, which in turn would pave the path for therapeutic interventions where cytokinesis plays an important role, such as polyploidy in certain cancers, and neurological diseases (Box 1).

## Figures and Tables

**Figure 1 cells-11-03977-f001:**
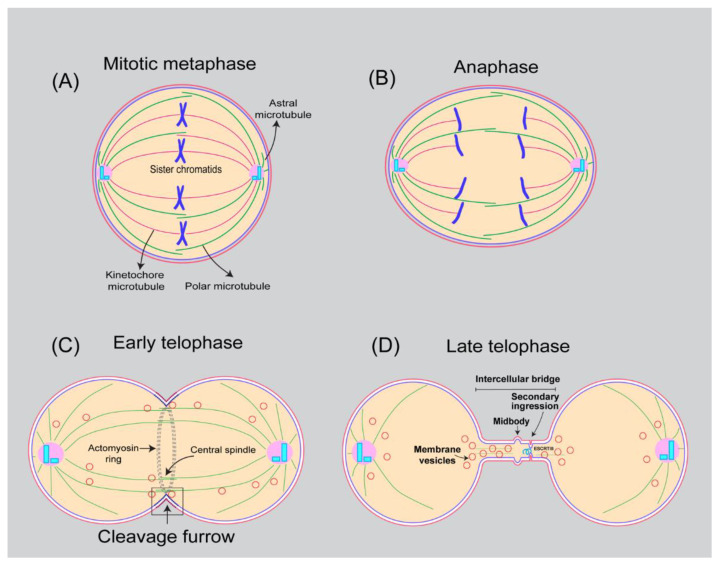
Schematic representation of mitotic cells undergoing division. (**A**) At metaphase, cells acquire round shape, chromosomes attached to kinetochore microtubules become aligned at the equatorial plane. (**B**) At anaphase, cells elongate, sister chromatids separate, and the daughter chromosomes move towards the poles. (**C**) Once daughter chromosomes reach poles and form two nuclei, the cytoplasm between the two nuclei is divided and physically separated by a process known as cytokinesis. During early stages of telophase, the mitotic spindle determines the site of cleavage furrow, by recruiting two master regulators Centralspindlin complex and Chromosomal Passenger Complex (CPC). These multisubunit complexes activate downstream signaling proteins including Rho-GEF (Ect-2), Rho GTPase and cytoskeletal regulators to promote assembly of actomyosin contractile ring (composed of unbranched actin and myosin-II) positioned midway between segregated chromosomes. Plasma membrane at the cytokinetic furrow is anchored to the contractile ring via Anillin and Septins and anionic phospholipid PI(4,5)P_2_. Constriction of the ring with concomitant membrane addition via both endocytosis and forward trafficking, generates the cleavage furrow. (**D**) Cleavage furrow ingression and stabilization results in the formation of intercellular bridge with an electron dense structure in the middle called midbody. The final steps of cytokinesis known as abscission involve endocytic trafficking, clearance of PIP2 on furrow membrane, microtubule severing, F-actin clearance, fusion of endosomes with the plasma membrane leading to secondary ingression. In a final step, there is assembly of the endosomal sorting complex required for transport (ESCRT III) helical filaments downstream of Cep55 and ALIX-TSG101 activation, resulting in membrane cutting that will physically separate the cytoplasm between two daughter cells.

**Figure 2 cells-11-03977-f002:**
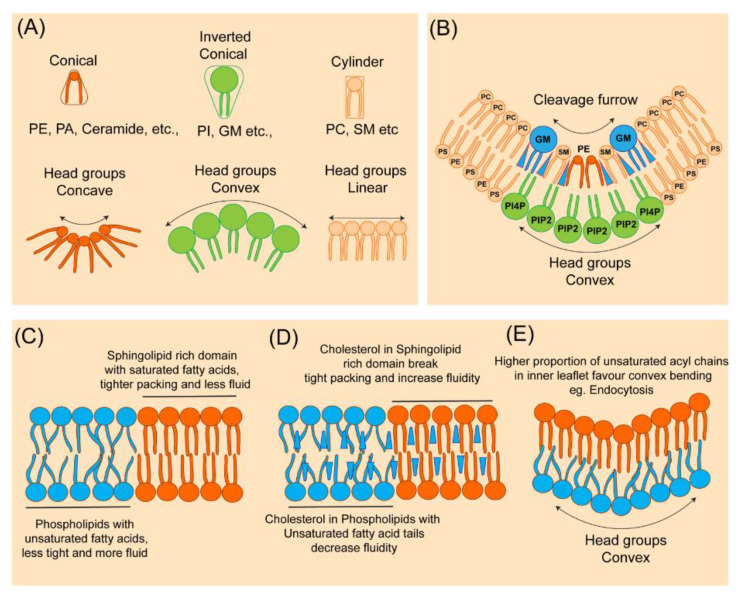
Chemical and physical properties of lipids and their influence on membrane curvature during cytokinesis. (**A**) Lipid shape is based on the cross-sectional area of the head group relative to their fatty acid acyl chains. (**B**) Representative lipid composition at the cleavage furrow of a typical mitotic cell. (**C**,**D**) Lipids with saturated fatty acids tend to pack more tightly than unsaturated fatty acid-containing lipids, and as a result, they are less flexible and rigid (**red**). However, when cholesterol is present, it can prevent the tight packing of saturated fatty acid-containing lipids and thus increase fluidity. In contrast, cholesterol increases the packing density of unsaturated fatty acids and thus reduces fluidity in these lipid domains. (**E**) Unsaturated fatty acid-containing lipids in the inner leaflet of the plasma membrane favor convex bending, for instance in the case of endocytosis.

**Figure 3 cells-11-03977-f003:**
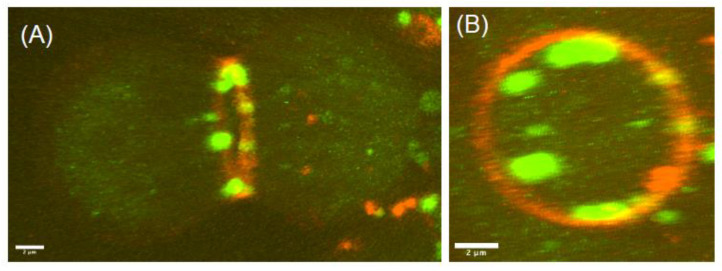
Multivesicular endosomes release their intraluminal vesicles in the cytosol closure to the cleavage. (**A**) Horizontal view of a spermatocyte undergoing cytokinesis, showing EYFP Rab7positive endosomes (**green**) docked onto the mRFP-Anillin ring (**red**). (**B**) A cross sectional view of a spermatocyte in (A), showing EYPF-Rab7 positive endosomes (**green**) docked onto the mRFP-Anillin ring (**red**).

**Figure 4 cells-11-03977-f004:**
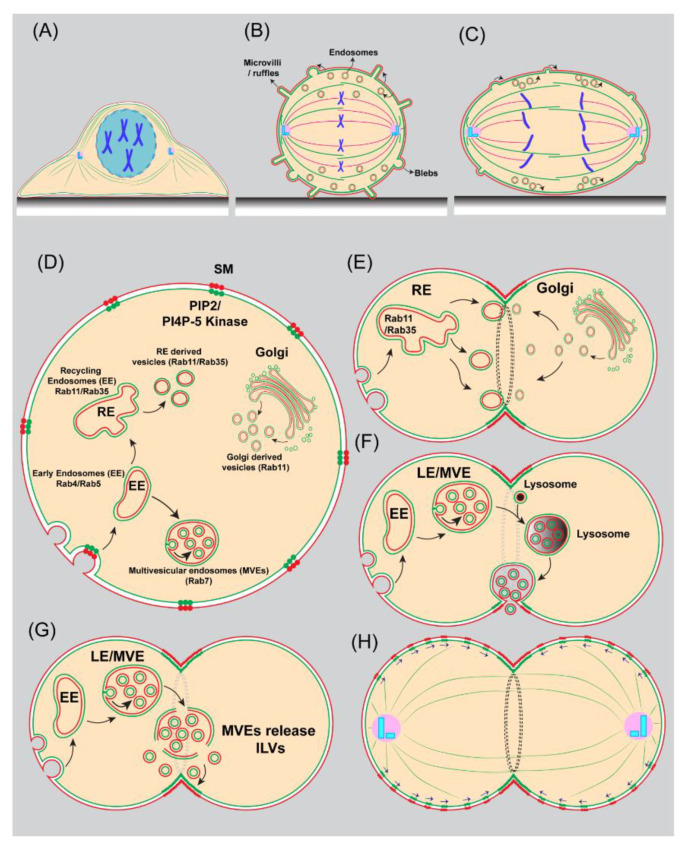
Mechanisms for membrane traffic and lipid polarization during cytokinesis. (**A**–**C**) Regulation of plasma membrane area in a dividing mammalian cell. (**A**) At the interphase, cultured, adherent mammalian cells appear flat; (**B**) at the onset of mitosis, cells lose their attachment to the substrate and begin to round up and reduce their surface area. To accommodate this change in surface area, the plasma membrane folds into blebs, microvilli, and ruffles and becomes endocytosed. (**C**) During anaphase, cell surface area increases, which correlates with the fusion of recycling endosomes with the plasma membrane and the expansion of membrane projections. (**D**) An eukaryotic cell in prophase showing endomembrane compartments, including early endosomes (EE), recycling endosomes (RE), multivesicular endosomes (MVEs), the Golgi apparatus, and vesicles derived from these compartments. The plasma membrane bilayer is shown with an outer leaflet (red) and an inner leaflet (green). Circular dots on the plasma membrane correspond to lipid domains, including sphingomyelin-rich domains (red dots) and Phosphatidylinositol-4, 5-bisphosphate (PIP2) and its biosynthetic enzyme PI(4)P 5-kinase (green dots). (**E**) A diagram depicting membrane trafficking from recycling endosomes and the Golgi apparatus to the cleavage furrow. (**F**) Cartoon showing membrane addition via lysosome exocytosis at the cleavage furrow in mammalian cells. Late endosomes (LE), Multivesicular endosomes (MVE). (**G**) Schematic representation of membrane trafficking via multivesicular endosomes that release their intraluminal vesicles in the cytosol near cleavage furrow in *Drosophila* spermatocytes. (**H**) Illustration of lateral movement of transbilayer-coupled lipid domains towards the cleavage furrow via a cortical flow mechanism.

**Table 1 cells-11-03977-t001:** Membrane/Lipid binding proteins involved in cytokinesis.

Protein Name/Protein Complex	Localization	Predicted/Known Lipid Binding	Mechanism in Cytokinesis	Ref.
ADP-ribosylation factor like protein 2–3 (ARL2, ARL3)	Midbody	N-terminal amphipathic helix/Anionic PLs.	Small GTPase, regulate microtubule dynamics	[170,171]
ADP-ribosylation factors (Arf1, Arf3, Arf6)	PM, Golgi, midbody, and endosomes	N-myristoylated	Small GTPases involved in Intra cellular vesicular trafficking, required during furrow ingression and abscission	[172,173]
Alix	Endosomes, midbody	Lysobisphosphatidic acid	Recruits ESCRT machinery to abscission site at midbody	[174,175]
ANCHR (abscission/NoCut checkpoint regulator; ZFYVE19)	Midbody	PIPs	Regulates the abscission checkpoint via retention of ESCRT component VPS4 at the midbody ring	[176]
Anillin	Contractile ring	PI(4,5)P_2_	Scaffolds contractile ring at the cell equator	[177,178]
Annexin A2; Annexin11	Cleavage furrow (A2), midbody (A11)	PIPs	AnnexinA2: connects equatorial cortex to central spindle and helps in localization of RhoGEF Ect2; Annexin A11, required for MKLP1 and Aurora B localization to the midbody	[179,180]
ARHGAP19	Cleavage furrow	Anionic PLs	Controls cytokinesis in T lymphocytes by acting as GAP for RhoA	[181]
Armadillo protein p007/Plakophilin-4 (PKP4)	Midzone, midbody	Unknown	Interact with RhoA and Ect2 and regulates Rho signaling	[182]
Capping protein (CAPZB)	Cleavage furrow	Anionic PLs	Required for midbody maturation, regulates actin dynamics	[183,184]
Chronophin/PDXP	Localizes to PM, Cleavage furrow, midbody	Unknown	Regulates cofilin dependent actin dynamics	[185]
Citron Rho-interacting kinase (CIT)	Cleavage furrow and midbody	Unknown	Regulates midbody formation	[186]
Cofilin-1	Cleavage furrow and midbody	PI(4,5,)P_2_	Regulate actin filament severing	[187,188]
ESCRT complex	Midbody	Anionic PLs	Involved in membrane cut during abscission	[189,190,191]
Exocyst complex Sec3, Sec5, Sec6, Sec8, Sec10, Sec15, Exo70 and Exo84	Endosomes, Early, late cleavage furrow and midbody	Sec3 and Exo70 interact PI(4,5)P_2_	Tethering of secretory vesicles to plasma membrane	[192,193,194,195]
Ezrin, Radixin, Moesin (ERM) proteins	Cholesterol dependent localization to cleavage furrow	PI(4,5,)P_2_	Regulate membrane to cytoskeleton interaction	[196,197]
F-BAR domain/Cdc15	Cell middle/contractile ring	Anionic PLs	Interacts with formin Cdc12 and promote contractile ring formation	[65]
Gin4 (Nim1 protein kinase)		PIPs	Septin assembly	[198]
Golgi phosphoprotein 3 (GOLPH3)	Golgi, cleavage furrow	PI4(P) effector	Required for localization of Rab11-positive PI(4)P enriched vesicles at the cleavage furrow	[199]
GPCRs	Spindle pole, midzone, midbody	Multi-pass membrane proteins	Affects actin cytoskeleton, knockdown causes defects	[200,201,202]
Integrin beta-1	Rab21 endosomes	Single pass membrane protein	Rab21 mediated integrin trafficking to and from the cleavage furrow	[203]
IPIP27	Endomembranes	PIPs	Scaffolds OCRL and couples it to endocytic BAR domain proteins (SH3PX1 or Pacsin2); involved in PI(4,5,)P_2_ homeostasis	[204]
MICAL (1 and 3)	Intercellular bridge	Unknown	MICAL-L1 and MICAL3 mediate targeting of Rab11-FIP3/Rab35 and Rab8 positive endosomes respectively to the ICB; regulate F-actin levels at midbody	[3]
MIT domain containing protein 1 (MITD1)	Late endosomes and midbody	PI(4,5,)P_2_	Interacts with ESCRT-III components and mediates abscission	[205]
Mso1 (Mint1) and Sec1 (Munc18)	Cell division site	Interact membrane via SNAREs	Vesicle fusion and cargo delivery; CR constriction, disassembly, and membrane closure defects	[206]
Myosin 19 (MYO19)	Mitochondria outer membrane	PA, PIP, PIP2 and PIP3	Involved in mitochondrial segregation via actin-based motor activity	[207,208]
Nonmuscle myosin-II	Cleavage furrow/ Contractile ring	Anionic phospholipids	Directly binds to membranes independent of F-actin	[209]
OCRL	Endosomes/intercellular bridge	PI(4,5)P_2_	Rab35 dependent localization to ICB, hydrolyze PI(4,5)P_2_ which in turn help in actin clearance.	[23]
Opy1	Dual PH domain containing protein	PI(4,5)P_2_	Endogenous PI(4,5)P_2_ sensor binds to its3	[210]
P190RhoGAP (ARHGAP35 and ARHGAP5)	Cleavage furrow	Anionic PLs	Regulates RhoA activity	[211,212]
P50RhoGAP (ARHGAP1)	Midbody	Anionic PLs	Promote actin clearance in the intercellular bridge	[213]
PDZD-8, TEX-2, OCRL and UNC-26/synaptojanin	ER/intracellular membrane compartments	PI(4,5)P_2_	Endosomal PI(4,5)P_2_ homeostasis	[214]
PI3K-C2α	Midbody	PI(3,4)P_2_	VPS36 binds to PI(3,4)P_2_ at the midbody and recruits CHAMP4B to mediate abscission	[10]
PI3K-III kinase complex (VPS15, VPS34, Beclin1, UVRAG and BIF-1	Localizes to endosomes and midbody	PI, PIP, N-myristoylation (Vps15)	Phosphorylates PI to PI(3)P; required for abscission	[215,216]
Pkd2/polycystins	PM/cell equatorial plane	Transmembrane protein	Membrane stretch activated Ca+ influx	[217]
PRIP (phospholipase C (PLC)-related catalytically inactive protein)	Cleavage furrow	PI(4,5)P_2_	regulates phosphoinositide metabolism at cleavage furrow	[218]
Prostate androgen regulated protein (PAR)	Centrosomes, spindle midzone, midbody	Single pass membrane protein	Form complex with Aurora A, Survivin, Aurora B and INCENP and crease Aurora B kinase activity	[219]
Protein kinase C epsilon (PRKCE)	Late cytokinetic furrow	DAG	Required for RhoA inactivation and actomyosin clearance during abscission	[164]
PTEN	Cleavage furrow	PI(3,4,5)P_2_	Contributes to PI(4,5)P_2_ production at the furrow.	[93,100]
PTEN/dPLCXD	Endosomes	PI(4,5)P_2_	Novel PI(4,5)P_2_ phosphatase, overexpression rescues OCRL loss of function phenotypes	[220]
Rab1	Golgi, cleavage furrow	S-geranylgeranylations	Interact with GOLPH3 and regulates vesicular trafficking	[221]
Rab10	Cleavage furrow midbody	S-geranylgeranylations	Unknown, possibly involved in delivery of Sorcin protein to cleavage site which is involved in regulating calcium homeostasis and Polo-like kinase-1	[222]
Rab14	Cleavage furrow/midbody	S-geranylgeranylations	Regulate actin clearance at the midbody.	[223,224]
Rab21	Early endosomes, cleavage furrow, midbody	S-geranylgeranylations	Integrin bet-1 trafficking during late steps of cytokinesis	[203]
Rab24	Mitotic spindle, cleavage furrow and intercellular bridge	S-geranylgeranylations	Affects kinetochore-microtubule attachment	[225,226]
Rab35	Recycling endosomes, cleavage furrow, mid body	PIP2, S-geranylgeranylations	PIP2 and F-acting remodeling during late steps of cytokinesis	[23,92,173]
Rab7	Late endosomes, multivesicular bodies	Small GTPase, S-geranylgeranylation	Membrane delivery to cleavage furrow	[131]
Rab8	Midbody	S-geranylgeranylations	Required for abscission	[227]
RacGTPase-activating protein-1 (MgcRacGAP)	Cleavage furrow and midbody	Anionic phospholipids	Regulates Rho GTPase	[112]
RalA and RalB	RalA (cleavage furrow) RalB midbody	Polybasic motif and S-geranylgeranylation	RalA (exocyst targeting to cleavage furrow); RalB (exocyst targeting to midbody during abscission)	[228,229,230]
Ras-related protein Rab11A and Rab11 family interacting protein 3 and 4	Recycling endosomes, Cleavage furrow midbody	Rab11 (small GTPase) S-geranylgeranylation	Endocytic traffic during late steps of cytokinesis,	[231,232]
Rho GTPase-activating protein 35 (ARHGAP35)	Cleavage furrow	Anionic phospholipids	Regulates actomyosin contractility	[211,212,233]
Rho GTPases (RhoA, RhoB, RhoC, Cdc42 and Rac1)	Cleavage furrow	Polybasic motif, S-geranylgeranylation (all), S-palmitoylation (Cdc42, RhoB) and S-farnesylation (RhoB)	Acting cytoskeleton organization, Rho is involved in cleavage furrow formation and ingression	[234]
Rho-associated protein kinases 1 and 2	Midbody	PIP2, PIP3, sphingosine, arachidonic acid, PC	Rho effector kinases. Regulates contraction of actomyosin ring by phosphorylation of myosin light chain	[235,236]
RhoGEF ECT2	Cleavage furrow, midbody	Phosphoinositides	RhoA activation and cleavage furrow formation	[105]
Septins (SEPT1-12, SEPT14)	Cleavage furrow, midbody	PI(4,5,)P_2_	GTPase, form filaments, rings during cleavage furrow ingression and abscission at midbody; act as diffusional barrier	[237,238]
Serine/threonine-protein kinase N2 (PKN2)	Cleavage furrow and midbody	C2 domain, anionic PLs, arachidonic acid	Function as Rho/Rac effector, required for abscission	[239,240]
Serologically defined colon cancer antigen (SDCCAG3)	Endosomes, midbody	unknown	Regulate Arf-mediated vesicular trafficking/signaling	[241]
SNARE (t-SNAREs include Sso, Sec9,) (v-SNARE Snc1/2)	Target SNAREs(t-SNARE/syntaxin) Vesicular SNARE (v-SNARE/VAMP)	Transmembrane proteins	Required for vesicle fusion at the target membranes	[3,242]
Sorting nexins (SNX9, SNX18, SNX33)	PX and BAR domain	PIP, PIP2, PIP3	Required for endocytosis dependent and independent roles in cytokinesis	[243]
Spartin (SPG20)	Lipid droplet and endosomes	unknown	Localizes to midbody by interacting with ESCRT-III protein Ist1	[244]
Spastin (SPAST)	ER, endosomes, midbody	unknown	Involved in microtubule severing during abscission	[245]
Spectrin beta chain	PH domain	PIP2, PIP3, PS	Lateral membrane biogenesis in concert with Ankyrin-G	[246]
STAM-Binding protein	Endosomes, cleavage furrow, midbody	unknown	Regulation of ubiquitylation at central spindle	[247]
Stomatin	Plasma membrane	Cholesterol	Change in lipid species including ether lipids and PC	[248,249]
Supervillin	Cleavage furrow midbody	Interact with lipid raft proteins	Links plasma membrane to the cytoskeletion	[250,251,252]
Ubiquitin carboxy-terminal hydrolase 8 (USP8)	Endosomes, midbody	Farnesylation ?	Regulation of ubiquitylation at central spindle	[247]
UNC119a	Dynamic localization; spindle midzone and midbody	Interact with N-myristoylated proteins	Has role in Fyn signaling and Rab11 dependent phosphorylation at midbody	[253]
Ync13/UNC-13/Munc13	Plasma membrane, cell tip, division site	PS, PIP2	Coordinates exocytosis, endocytosis, and cell wall integrity	[254]
YPP1 and Efr3	Plasma membrane	Basic residues in Efr3 binds to anionic PLs.	Scaffolds Stt4 (PI4KIIIalpha)	[255,256,257]
Zinc-finger FYVE domain-containing protein 26 (ZFYVE26)	midbody	PI(3)P	Interact with ESCRT-III subunits to mediate abscission	[258]

## Data Availability

Not applicable.

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
