# Peer review of "Lipid Polarization during Cytokinesis"

_cells, 2022, doi:10.3390/cells11243977_

Round 1

Reviewer 1 Report

The manuscript entitled “Lipid polarization during cytokinesis” summarises past and recent data regarding the role of lipids in cytokinesis. In my opinion, the review is well written and structured and complete with several figures that facilitate the reading.

Few suggestions to improve the manuscript:

1.       Citing alterations in cytokinesis occurring in sphingolipidoses or other disorders could be of interest (for example the case of beta-glucosylsphingosine for endothelial cytokinesis and psychosine for globoid cell formation)

Reviewer 2 Report

In this very detailed, well referenced, and timely review, Kunduri et al discuss the lipid polarisation and organisation in the cell surface during cytokinesis. The importance of membrane lipid organisation and signaling is often poorly appreciated in the cytoskeleton-heavy papers of cell-division field, and this review will stand out as a valuable resource in the field literature. Authors have done a terrific job of highlighting the physiological roles of different lipid species in the PM and how they localise and influence cytokinesis. The schematics are high-quality and already publication quality. I have some additional comments which I believe will contribute to the further general appeal of this review.

Major comments:

1) Please discuss in a bit more details how cortical actin cytoskeleton interactions (and flows) with plasma membrane (PM) influence formation of lipid domains. Authors have already hinted this in different sections (for eg. line 110-114 on ‘lipid rafts’ and line 517-520 on ‘cortical flows’). However, these two seemingly different processes are connected based on the emerging results from work of a few eminent labs in the membrane field, as suggested in several recent papers (PMID suggested below). Authors should briefly discuss that the lateral segregation of lipids at PM emerges from the chemical (like lipid tail length and saturation), geometric features (like shape) of the lipid and the dynamics of  the cortical F-actin cytoskeleton that associates with the inner-leaflet. Authors have done a good job of citing pioneer studies from Kai Simons and colleagues on ‘lipid rafts’ but they should also include citations from Ken Jacobson (PMID: 31051105 ) and Akihiro Kusumi (PMID: 22905956) in support of the idea of the ‘picket-fence model’ which postulated that lipid diffusion can be corralled by the juxtamembrane cortex. The cytokinetic furrow is intimately connected with the underlying contractile cortex and it’s very likely the lipid organisation in these regions arise from a combination of behaviors expected form both ‘picket-fence’ and ‘lipid-raft’ model.

2) On similar notes, authors have discussed the transbilayer asymmetry seen in eukaryotic cells but haven’t discussed the potential consequence of this arrangement on the nano to mesoscale organisation of the bilayer. Recent work from Jitu Mayor group among others have shown in a series of critical studies, the dynamic actin cortex associates with the inner-leaflet PS lipids which engage in transbilayer coupling with outer-leaflet to regulate the formation of F-actin and cholesterol-sphingolipid dependent lipid-order domains (PMID: 22682254, 25910209, 35867835). Specifically, the emerging idea of ‘active emulsions’ as they discuss in their very recent paper (PMID: 2022) could be incorporated along the ideas of lo-ld phase separation in presence of cholesterol and sphingolipid (Like Fig 2C and related text). Please include these references to buttress the point that the asymmetric transbilayer coupling between leaflet is a key organizing principle for domains on the cell surface. This links well with both lipid polarisation and cortical flow concepts presented in the manuscript.

Minor Comments:

- Please address the typographical errors

- What is Figure 7 in line 324 ?

Reviewer 3 Report

Comments for Kunduri et al

This is a large review of the lipid composition and cytokinesis. It looks to be a good review, and I would suggest several points.

The focus of this review is the lipid itself, but the proteins involved are described. The table of the membrane or lipid binding proteins involved in cytokinesis will help the readers.

The BAR domain protein Cdc15 is omitted but would be the important membrane-binding protein for cytokinesis.

The section starting with Membrane trafficking… would be the summary for the following sections, but the bullet points are not corresponding to the following sections. The subheads should be consistent with the points i)- iv) in the section. The reservoir for membrane supply will be important, but the illustration in Figure 4 does not cover all of these 4 points.

The writing needs more sophistication. The comma is often omitted.

Round 2

Reviewer 2 Report

Authors have made significant revisions in response to the feedback from reviewers. The revised version will be a valuable addition to field literature.